



# Quantification of $CO_2$ hotspot emissions from OCO-3 SAM $CO_2$ satellite images using deep learning methods

Joffrey Dumont Le Brazidec[1], Pierre Vanderbecken[1], Alban Farchi[1], Grégoire Broquet[2], Gerrit Kuhlmann[3], and Marc Bocquet[1]

[1]CEREA, École des Ponts and EDF R&D, Île-de-France, France
[2]Laboratoire des Sciences du Climat et de l'Environnement, LSCE/IPSL, CEA-CNRS-UVSQ, Université Paris-Saclay, 91198 Gif-sur-Yvette, France
[3]Empa, Laboratory for Air Pollution / Environmental Technology, Dübendorf, Switzerland

**Correspondence:** Joffrey Dumont Le Brazidec - Current affiliation: European Centre for Medium-Range Weather Forecasts (ECMWF), Bonn, Germany (joffrey.dumont@ecmwf.int)

**Abstract.**

This paper presents the development and application of a deep learning-based method for inverting $CO_2$ atmospheric plumes from power plants using satellite imagery of the $CO_2$ total column mixing ratios ($XCO_2$). We present an end-to-end CNN approach, processing the satellite $XCO_2$ images to derive estimates of the power plant emissions, that is resilient to missing data in the images due to clouds or to the partial view of the plume due to the limited extent of the satellite swath.

The CNN is trained and validated exclusively on $CO_2$ simulations from 8 power plants in Germany in 2015. The evaluation on this synthetic dataset shows an excellent CNN performance with relative errors close to 20%, which is only significantly affected by substantial cloud cover. The method is then applied to 39 images of the $XCO_2$ plumes from 9 power plants, acquired by the Orbiting Carbon Observatory-3 Snapshot Area Maps (OCO3-SAMs), and the predictions are compared to average annual reported emissions. The results are very promising, showing a relative difference of the predictions to reported emissions only slightly higher than the relative error diagnosed from the experiments with synthetic images. Furthermore, the analysis of the area of the images in which the CNN-based inversion extract the information for the quantification of the emissions, based on integrated gradient techniques, demonstrates that the CNN effectively identifies the location of the plumes in the OCO-3 SAM images. This study demonstrates the feasibility of applying neural networks that have been trained on synthetic datasets for the inversion of atmospheric plumes in real satellite imagery of $XCO_2$, and provides the tools for future applications.

## 1 Introduction

The satellite imagery of total column average dry air mole fraction of carbon dioxide ($XCO_2$) from the Snapshot Area Map (SAM) mode of the Orbiting Carbon Observatory-3 (OCO-3) (Eldering et al., 2019) or the forthcoming CO2M mission (Janssens-Maenhout et al., 2020; Meijer et al., 2023) are pivotal for monitoring carbon dioxide ($CO_2$) emissions. In the vicinity of large $CO_2$ anthropogenic sources, such as power plants, the satellite images may include $CO_2$ atmospheric plumes emanat-





ing from these sources. From these images, atmospheric inversion approaches can estimate the $CO_2$ emissions of the sources by analysing the signal intensity of the detected plumes (Nassar et al., 2017; Reuter et al., 2019; Chevallier et al., 2019; Wu et al., 2020; Zheng et al., 2020; Nassar et al., 2022; Chevallier et al., 2022; Cusworth et al., 2021).

Various approaches can be used to determine the emissions underlying the $XCO_2$ plumes in the satellite imagery. A first category rely on traditional atmospheric inversion methods that minimise the misfits between the satellite observations and simulations of the plumes with relatively expensive (Eulerian or Lagrangian) transport models to identify the optimal emission estimate (Pillai et al., 2016; Broquet et al., 2018). A second category are light-weight methods that apply the principle of mass conservation to compute the emissions from the $CO_2$ enhancement of the emission plume (such as integrated mass

enhancements, divergence methods, and cross-sectional flux methods) or compare the observed plume with a Gaussian plume model (Gaussian plume inversions). Light-weight methods rely on wind fields taken, e.g., from meteorological reanalysis products. These light methods have been evaluated in several studies (e.g., Varon et al., 2018; Hakkarainen et al., 2024; Danjou et al., 2023; Santaren et al., 2024; Kuhlmann et al., 2024; Danjou et al., 2024). Despite the advancements in $CO_2$ plume inversion techniques, significant challenges remain, notably: 1) the extraction of plumes from $XCO_2$ backgrounds, hindered

by low signal-to-noise ratios due to the large amplitude of background variations associated to the $CO_2$ natural fluxes, and to relatively large noise in the image (due to instrumental errors and to uncertainties in the retrieval of mole fractions from the satellite measurements), (2) the complex process of deducing the source emissions from clearly delineated plumes, marred by uncertainties in the corresponding transport and dispersion (i.e., in either the transport modelling or in the wind field and assumptions regarding the vertical structure of the 3D $CO_2$ plume for the derivation of the effective wind driving the 2D $XCO_2$

plume in the light-weight analysis (Dumont Le Brazidec et al., 2022), and (3) reconstructing emissions from images with a partial view of the plumes due to missing data where there are clouds or gaps in satellite coverage.

Machine learning models have been suggested in response to these obstacles and have been primarily applied to $CH_4$ and $NO_2$ images (e.g., Lary et al., 2016; Finch et al., 2021; Jongaramrungruang et al., 2021; Joyce et al., 2023; Kumar et al., 2023). Our previous work (Dumont Le Brazidec et al., 2022, 2023) pioneered the use of deep learning methodologies, specifically

Convolutional Neural Networks (CNNs), for the segmentation and inversion of $CO_2$ plumes for the estimate of point sources. This approach has demonstrated its efficacy in addressing these challenges when tackling synthetic satellite image with a full coverage of the plumes, i.e. without the loss of observations due to cloud cover or quality control in the satellite limited field of view. This paper is a direct continuation of Dumont Le Brazidec et al. (2023). Specifically, our approach involves developing a supervised learning CNN system designed to predict $CO_2$ emissions using $XCO_2$ images and ancillary data (such as wind

fields, time, and $NO_2$ images which will be measured by CO2M). This CNN is trained on a synthetic dataset, constructed from model simulations, comprising synthetic $XCO_2$ fields and the corresponding true emissions. Through this training process, the CNN learns to correlate specific features within the input images covering the plume from a targeted point source with certain output values, namely the emissions from the point source. The CNN's capability to generalise is subsequently assessed using a new, unseen dataset during the training phase. In particular, this assessment is based on tests targeting a source which was

not covered by the synthetic images used for the training phase.





In previous studies, the models have only been tested with synthetic images without missing data. In this study, we advance our methodology to quantify the emissions from real satellite images, specifically 39 OCO-3 SAMs. Those images of $64km^2$ cover 9 power plants located in the USA (7), Europe (1), and China (1). To make this possible, this paper introduces a new upgrade of the CNN approach to address the third principal challenge in $CO_2$ plume inversion: handling images with a partial cover of the plumes due to the loss of observations associated to clouds, or due to the limited extent of the satellite swath. Furthermore, the training of the CNN involves a novel data augmentation strategy, specifically the incorporation of beta or uniform distribution mappings for plumes and the corresponding emissions. This enhancement aims to improve the robustness and stability of the CNN in predicting $CO_2$ emissions under various conditions.

The structure of this paper is as follows: Section 2 introduces the synthetic dataset, which bears significant resemblance to the one described by Dumont Le Brazidec et al. (2023) and Santaren et al. (2024), the OCO-3 SAMs utilised exclusively for evaluation, and the dataset's training/validation/test split strategy. Section 3 details the model, the developed data augmentation approach aimed at stabilising CNN training, the methodology for addressing the problem of clouds, and the training parameterisation. In Section 4, we successively present the CNN's emission estimations for plumes across the synthetic and OCO-3 SAM datasets. Special attention is given to analysing the model's OCO-3 SAM predictions through the lens of integrated gradients, a method that elucidates the contribution of each input feature to the model's predictions, enhancing interpretability. Finally, Section 5 discusses cogent future directions, before we conclude.

## 2 Dataset

### 2.1 Synthetic dataset

The synthetic dataset employed in this study is very similar to the one used by Dumont Le Brazidec et al. (2023). The dataset consists of hourly $XCO_2$ and $NO_2$ fields from the SMARTCARB project (Brunner et al., 2019; Kuhlmann et al., 2019), which generated one year of synthetic CO2M observations from high-resolution $CO_2$ and $NO_2$ transport simulations covering power plants in Germany, Poland and the Czech Republic. The SMARTCARB dataset has been used in various studies for assessing emission quantification methods (e.g., Kuhlmann et al., 2020, 2021; Hakkarainen et al., 2021; Santaren et al., 2024). For this study, we extracted 32×32 pixel (2km resolution) fields centred around different power plants. For comparison, in Dumont Le Brazidec et al. (2023), the image size was chosen as 64×64. The transition to focusing analysis on a more confined area surrounding the hotspots (power plants) is driven by several factors: i) the critical portion of the plume influencing emission reconstruction typically lies within this central area, as noted in Dumont Le Brazidec et al. (2023), ii) satellite swath constraints, and iii) this more focused approach demonstrates a stabilising effect on neural network training, likely due to the reduction of superfluous information.

To account for the inherent noise of satellite instruments, we introduce Gaussian random noise with a standard deviation of $0.7$ ppm to the $XCO_2$ images, reflecting typical noise levels expected for OCO-3 and $CO_2$M snapshots as reported by Meijer (2020); Taylor et al. (2023); Danjou et al. (2024). Given the observed strong correlation between $NO_2$ and $CO_2$ plumes and





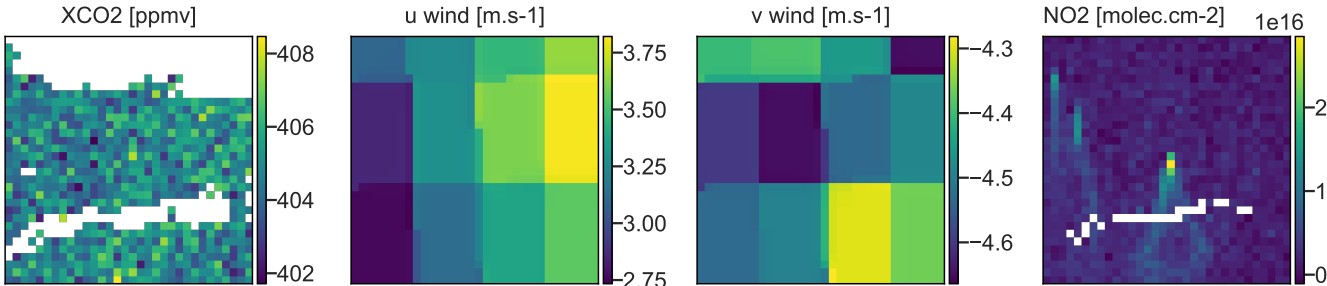

**Figure 1.** Examples of inputs used by the CNN model. The first, second, third and fourth columns represent the $XCO_2$ images, vertically averaged $u$ and $v$ winds, and $NO_2$ images, respectively. The power plant of interest is always located in the middle of the image.

$CO_2$M's capability to measure $NO_2$, we incorporate noisy $NO_2$ fields in our analysis, characterised by Gaussian noise with a variance of $1e15\,\text{molec.cm}^{-2}$ (Kuhlmann et al., 2019).

Similarly to Dumont Le Brazidec et al. (2023), we integrate ERA5 wind data as additional inputs to the CNN model, aligning its original resolution of $28\,\text{km}$ with the $2\,\text{km}$ resolution used for the $CO_2$ and $NO_2$ images. Specifically, we employ 2D $u$ and $v$ wind fields, representing the average zonal and meridional winds, respectively, across the five lowest model levels of ERA5. This averaging process approximates the atmospheric conditions below $100\,\text{m}$.

To include the impact of cloud cover in the inversion of $CO_2$ plumes, we use the simulated cloud cover fractions extracted from the SMARTCARB dataset to mask pixels where retrievals are not available due to high cloud fraction. Following Kuhlmann et al. (2019), we use a cloud threshold of 1% for $CO_2$ images and 30% for $NO_2$ images.

Moreover, we study the interest of introducing temporal information to our CNN inputs, by incorporating the hour of the day, day of the week, and day of the year. To capture the cyclical nature of time, these features are transformed into cosine and sine representations, ensuring proximity between temporally adjacent data points (e.g., the last and first hours of the day). Consequently, each $XCO_2$ field is associated with a vector of six scalar values encoding the temporal context of the observation. In Fig. 1, we present a typical input data used by the CNN to predict the emissions of the local hotspot.

## 2.2 Set of OCO-3 SAMs

The OCO-3 SAM mode is an observation strategy designed to monitor $CO_2$ emissions from specific emission hotspots (large urban areas and large industrial point sources). Unlike its standard observation mode, which conducts continuous scans of the Earth's atmosphere in nadir or glint mode, the SAM mode is a targeting mode, which provides high-resolution $XCO_2$ images around such emission hotspots. In this study, we have selected 39 OCO-3 SAMs at nine power plants to evaluate the applicability and reliability of our CNN model trained on synthetic datasets. We selected OCO-3 SAM images corresponding to power plants for which reports of the emissions are available and have been studied in the scientific literature, and to a sufficient number of cloud-free $XCO_2$ retrievals of good quality. The list of power plants selected are described with average reported emission and number of collected SAMs in Table 1.

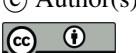



| Power plant | Coordinates [°N, °E] | Reported emissions [Mt CO$_2$/a] | Number of OCO-3 images |
|---|---|---|---|
| Colstrip | [45.88, -106.61] | 13.6 | 4 |
| Belchatow | [51.27, 19.33] | 37.6 | 4 |
| Tuoketuo | [40.20, 111.36] | 29.5 | 8 |
| Cumberland | [36.39, -87.65] | 12.4 | 3 |
| Labadie | [38.56, -90.84] | 15.0 | 6 |
| Intermountain | [39.51, -112.58] | 5.0 | 2 |
| Hunters | [39.17, -111.03] | 7.2 | 7 |
| Parish | [29.48, -95.63] | 13.2 | 4 |
| Conemaugh | [40.38, -79.06] | 16.9 | 1 |

**Table 1.** List of power plants selected for this study, along with their annual reported emissions, coordinates, and number of observations used. Emissions statistics are sourced by (Nassar et al., 2021; Grant et al., 2021; Lin et al., 2023) which are based on the US Environmental Protection Agency (EPA) (https://www.epa.gov/airmarkets/power-sector-emissions-data) and the European Pollutant Release and Transfer Register (E-PRTR, https://prtr.eea.europa.eu).

To adapt the raw OCO-3 SAM data for CNN analysis, we first construct a $32 \times 32$ grid with a resolution of 2 km (similar to the resolution of OCO3-SAM or CO2M) and centred at the power plant. Each grid cell is populated through a weighted interpolation of surrounding OCO-3 SAM data pixel centers, considering only those within a distance of less than 0.66 times the new grid resolution. This specific distance threshold was determined through experimentation to optimally preserve information

from the original dataset. Although this mapping strategy provides a straightforward means of converting OCO-3 SAM data into a format compatible with our CNN, it is acknowledged that this approach has limitations. Additionally, since most OCO3-SAM snapshots used in this study were taken in 2021 or 2022, the synthetic images are adjusted to account for the effect of climate change and the general increase of 2.3ppm per year in CO$_2$ concentration since 2015 (SMARTCARB synthetic dataset year). Figure 2 illustrates with 8 examples the process of transforming original OCO-3 SAM data into an XCO$_2$ field suitable

for CNN reconstruction.

### 2.3   Training, validation, and test split choices

For tests on synthetic and real data, a rigorous geographical separation is maintained between the power plants used in the training and validation datasets and those in the test dataset to avoid data leakage. For instance, when training a model to predict emissions from the Boxberg power plant, Boxberg plumes are excluded from the training set. The validation dataset

comprises plumes from a different power plant, Dolna Odra, which is neither used to train nor test the CNNs. This splitting strategy is exposed in Fig. 3.

This approach mirrors the strategy adopted in Dumont Le Brazidec et al. (2023) for analysis on synthetic images. We focus on the same three power plants for the tests: Lippendorf, Boxberg, and Turow, training distinct models for each to predict their

**Figure 2.** Examples of OCO-3 SAM observations for eight different power plant and their transformation into CNN-compatible images. For each case, on the left is described the original OCO-3 SAM data and on the right the corresponding CNN-compatible mapping (for which the power plant is always at the centre of the image). All values are in ppmv.

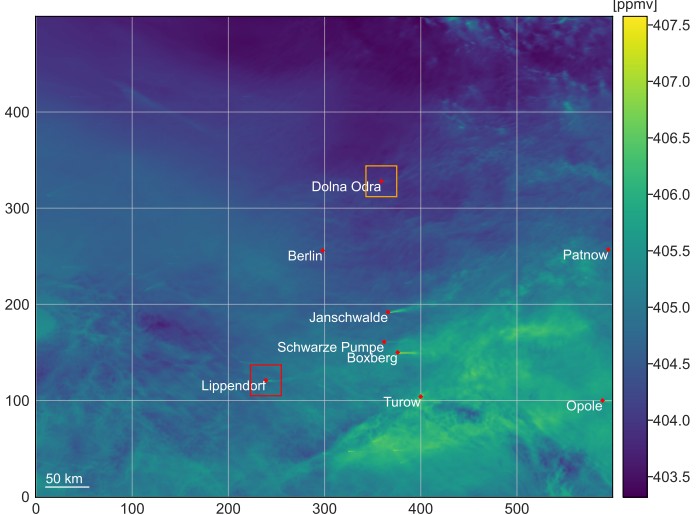

**Figure 3.** Map of XCO$_2$ concentrations within the complete SMARTCARB domain on 12 December at 03:00 UTC. The depicted XCO$_2$ fields are devoid of synthetic satellite noise for visibility. When constructing a model to predict emissions from Lippendorf, based on Lippendorf-centred fields (indicated by the red square), images from Dolna Odra (indicated by the orange square) are used for validation, whereas images from the remaining power plants serve as training data.

emissions. These models share the same architectural framework, hyperparameters, CNN structure, and preprocessing layers, but are trained on a dataset excluding plumes from its target power plant.

The rationale behind selecting Lippendorf, Boxberg, and Turow as test power plants is thoroughly discussed by Dumont Le Brazidec et al. (2023). Briefly, these power plants were chosen for their distinct characteristics: Lippendorf averaged emissions are equal to 15.2 Mt CO$_2$/yr; Boxberg's plume is often located close to other power plant plumes and its average emissions amount to 19.0 Mt CO$_2$/yr ; and Turow is characterised by low emissions of 8.7 Mt CO$_2$/yr. This selection criteria ensures an evaluation of the proposed CNN architecture across various emission scenarios.

It is critical to underline that while the test dataset for one experiment becomes part of the training dataset for another, each experiment was conducted independently, ensuring that model tuning was not optimised by outcomes derived from the test datasets. Finally, in our assessment of CNNs against OCO-3 SAM data, the training was based exclusively on synthetic data.

## 3 Deep learning methodology

The goal of this study is to determine the CO$_2$ emission rate (in $\mathrm{MtCO_2.yr^{-1}}$) of the hotspot in the center of a XCO$_2$ image using a CNN model, which takes the XCO$_2$ image alongside other data as input. This section describes the CNN model and





the data augmentation strategy, with a particular focus on the method to address cloud interference, and discusses training parameters.

## 3.1 CNN model and preprocessing layers

This subsection describes the CNN-based inversion system (the CNN model with its preprocessing layers that estimates emissions from images) and how it is trained. The CNN-based inversion system is a compound of preprocessing layers and a core CNN model. Preprocessing layers are operations successively applied to the $XCO_2$ fields and ancillary data before they are processed by the core CNN model. The core CNN model is a statistical model whose parameters (or neurons) are optimised during the training phase: its function is to identify and extract features from the input data, which it learns to associate with

specific levels of emissions. The training phase of the CNN-based inversion system consists in a series of five steps, depicted in Fig. 4 and in the following paragraphs.

### 3.1.1 Data augmentation

The data augmentation process creates an artificially infinite dataset from the SMARTCARB dataset to prevent the model from overfitting due to the SMARTCARB dataset's limitations. Specifically, instead of using XCO2 field directly from the

SMARTCARB dataset to train the core model, we use a composition of five different elements:

1. The principal component is a synthetic image centred on a major power plant of interest, exclusively containing the $XCO_2$ plume from that facility and the other major power plants. The SMARTCARB dataset composition facilitates isolating this field from all other anthropogenic and biogenic fluxes. This first component undergoes a distribution mapping: an emission level is randomly drawn from either a beta (to mitigate the training on extreme emissions) or a uniform

distribution as shown in Fig. 5, and the plume image is adjusted accordingly. Simultaneously, the CNN output is also adjusted at the emission level.

2. To this first component is added a randomly drawn $XCO_2$ background, which is augmented by summing it with a random number $b \sim U(-3.5, 3.5)$ (in ppmv), added uniformly across the field, in a manner analogous to Dumont Le Brazidec et al. (2023). The selection of the background (and all subsequently described elements) through uniform random draw-

ing is independent from the position of the main plume component.

3. To this is added other anthropogenic $XCO_2$ plumes identified in the SMARTCARB area, each scaled by a random factor ranging from 0.33 to 3.

4. The application of cloud cover constitutes the fourth component. A random selection of cloud cover from the SMART-CARB area is made, independent from the selections for other fields. $XCO_2$ pixels are deemed unobserved when cloud

cover exceeds 0.01, leading to replacement with NaN, which are subsequently replaced with the minimum value across all $XCO_2$ fields. This is meant for the CNN model to learn to ignore this non-informative constant value.

5. The fifth component adds to the other fields random Gaussian noise with a variance of 0.7 ppmv.

**Figure 4.** Description of the inversion system at training time as a compound of preprocessing layers and the model. The CNN-based inversion system consists in five steps: 1) construction of an XCO2 field as a sum of background XCO2, major and minor CO2 plumes, cloud cover and synthetic satellite noise, 2) concatenation with ancillary data (winds, time, and NO2), 3) standardisation of the fields, 4) processing by the core CNN model, and 5) and backpropagation.

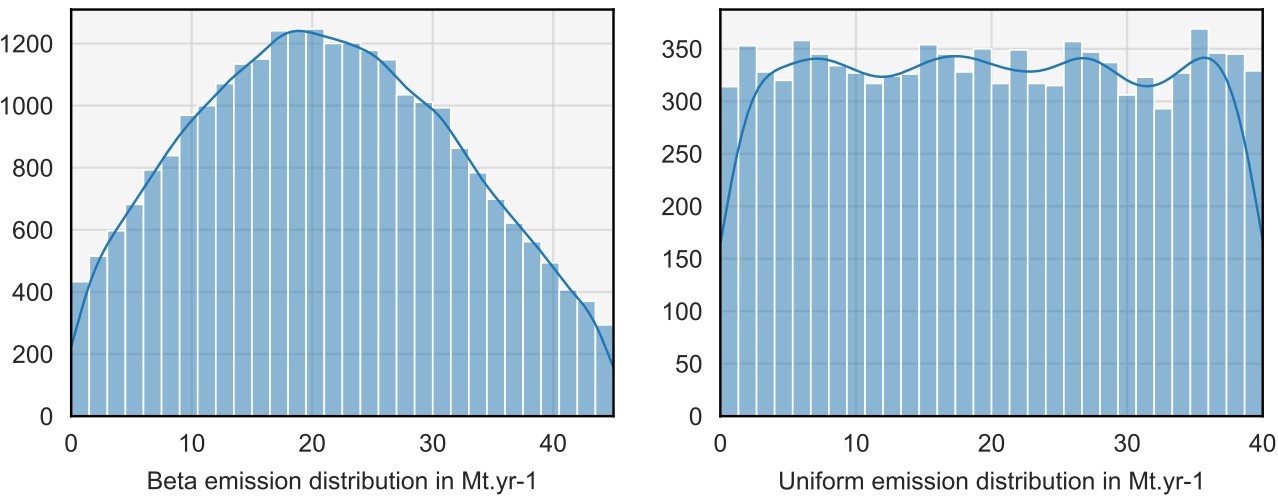

**Figure 5.** At training time, the hotspot emission and corresponding plume are adjusted based on a random draw from either a beta or uniform distribution.

### 3.1.2 Concatenation

The concatenation of the main $XCO_2$ field with the ancillary data represents the second step. The ancillary data may include wind conditions, the time and date of the observation, and the $NO_2$ field. In instances where the $NO_2$ field is incorporated, it also undergoes a data augmentation process (not depicted in Figure 4). Initially, the $NO_2$ plume is scaled by a random factor ranging from 0.75 to 2 to ensure the $NO_2$ plume amplitude are decorrelated from that of the $CO_2$ plume, thus preventing the core CNN model from relying for inversion on the tight correlation between the NOx and CO2 emissions. In principle, due to the large variations and uncertainties in the CO2-to-NOx emission and lifetime of NOx, $NO_2$ should primarily support the plume detection in the overall inversion process. Subsequently, the $NO_2$ field is partially masked due to cloud coverage. For this, we adopt the criterion from Kuhlmann et al. (2019) that an $NO_2$ pixel is marked as NaN if its cloud coverage fraction exceeds 0.3. Furthermore, the $NO_2$ field is subject to Gaussian noise with a variance of $1 \times 10^{15}$ molecules.cm$^{-2}$ (Kuhlmann et al., 2019).

### 3.1.3 Normalisation

Z-score normalisation of each physical field within the concatenated input data constitutes the third step, performed independently for each channel.





### 3.1.4 Processing

The fourth step is the core CNN model mapping from $XCO_2$ and ancillary fields to a scalar emission value. This model, consistent with the one described in Dumont Le Brazidec et al. (2023), features a series of convolutional, max pooling, batch normalisation, and dropout layers, with a total of $186,000$ trainable parameters. Specifically, if time and date features are used, they are integrated into the CNN post feature extraction (following the last dense layer).

### 3.1.5 Backpropagation

The final step entails computing the loss gradient, enabling neuron adjustments within the core CNN model through backpropagation.

In contrast to the training phase, the inversion system at evaluation phase consists only in the concatenation, normalisation, and processing by the core CNN model. The synthetic test dataset consists in pre-constructed, physically consistent simulated data (except for clouds as explained in Section 3.2), maintaining consistency with the methodology outlined in Dumont Le Brazidec et al. (2023).

### 3.2 Clouds

To assess the impact of cloud cover on CNN performance, we consider models trained and tested on varied datasets distinguished by varying degrees of fraction of cloudy pixels in the $XCO_2$ images:

- A first series of models are trained and tested on $XCO_2$ images under clear-sky conditions.

- A second series of models are trained and tested on $XCO_2$ images with cloud coverage ranging from $0\%$ to $25\%$.

- A third series of models are trained on $XCO_2$ images with cloud coverage from $0\%$ to $50\%$ but are tested on images with cloud coverage from $25\%$ to $50\%$.

- A final series of models are trained on $XCO_2$ images with cloud coverage from $0\%$ to $75\%$ but are tested on images with cloud coverage from $50\%$ to $75\%$.

These varying degrees of cloud coverage are constructed through random sampling of cloud coverage over the SMARTCARB domain. This method of training and testing models under varying cloud conditions allows us to compare the degradation in model performance with increased cloud coverage. Additionally, training the model tested on cloud coverage between $50\%$ and $75\%$ on a range from $0\%$ to $75\%$ ensures the maintenance of a "universal" model capable of inverting plumes in scenarios with both low and high cloud coverage.

### 3.3 Training parameterisation

We configure the training hyperparameters as follows: the model uses the Adam optimiser, with an initial learning rate of $1 \times 10^{-3}$, which is adjusted according to a reduce-on-plateau strategy down to $1 \times 10^{-5}$ with a patience parameter set to 20.





| Power plant | with NO2 | cloud fraction | | | |
|---|---|---|---|---|---|
| | | 0 | 0 to 25% | 25 to 50% | 50 to 75% |
| **Lippendorf** | ✗ | 23.1 | 22.8 | 25.1 | 23.7 |
| **Lippendorf** | ✓ | 17.9 | 18.7 | 21.6 | 23.5 |
| **Boxberg** | ✗ | 18.5 | 19.5 | 20.2 | 21.1 |
| **Boxberg** | ✓ | 15.6 | 15.5 | 16.7 | 19.5 |
| **Turow** | ✗ | 35.8 | 35.4 | 37.6 | 57.3 |
| **Turow** | ✓ | 24.8 | 28.9 | 37.9 | 65.2 |

**Table 2.** Median of the relative error between the CNN predictions and the true emissions for Lippendorf, Boxberg, Turow, for varying levels of clouds and input configurations.

The batch size is established at 128, and the training process spans 750 epochs. These parameters were selected based on a rigorous experimental process, combined with adherence to established practices in the field. For the loss function, the Mean Absolute Error (MAE) was chosen.

## 4 Application to synthetic and OCO-3 SAM observations

### 4.1 Application to synthetic dataset

Similarly to Dumont Le Brazidec et al. (2023), we investigate the performance of various CNN models in predicting the emissions of the Lippendorf, Turow, or Boxberg power plants. A collection of CNNs undergoes training on subsets of power plants, each excluding one for evaluation. For each power plant, a collection corresponds to models that are trained and tested on images affected by varying levels of cloud coverage. In addition, the models are trained with two different input configurations: one that includes $XCO_2$, wind, time, and $NO_2$ data, and another that includes all these variables except for $NO_2$. As a result, a total of 3 (number of target power plants) $\times 4$ (cloud coverage scenarios) $\times 2$ (input configurations) $= 24$ CNNs are trained and evaluated.

Figures 6 and 7 show Kernel Density Estimation (KDE) plots for the absolute relative error and the algebraic difference between the model predictions and the true emissions for the configuration without and with $NO_2$ input. A comprehensive summary of the results is also provided in Table 2.

The novel data augmentation strategy presented in Section 3.1 improves the stability of the performances of the CNNs in comparison to Dumont Le Brazidec et al. (2023), making it unnecessary to train a CNN ensemble to achieve satisfactory and consistent results. Significantly, in comparison to Dumont Le Brazidec et al. (2023), Boxberg median relative error with $NO_2$ decreased from 36.9% to 15.6%. Furthermore, (not shown) an improvement in the results is observed when using the mean of the emissions predicted by applying the CNN to an ensemble of images with added Gaussian noise. Specifically, for added Gaussian noise with a standard deviation of 0.3, the Lippendorf median relative error (without additional inputs) decreases

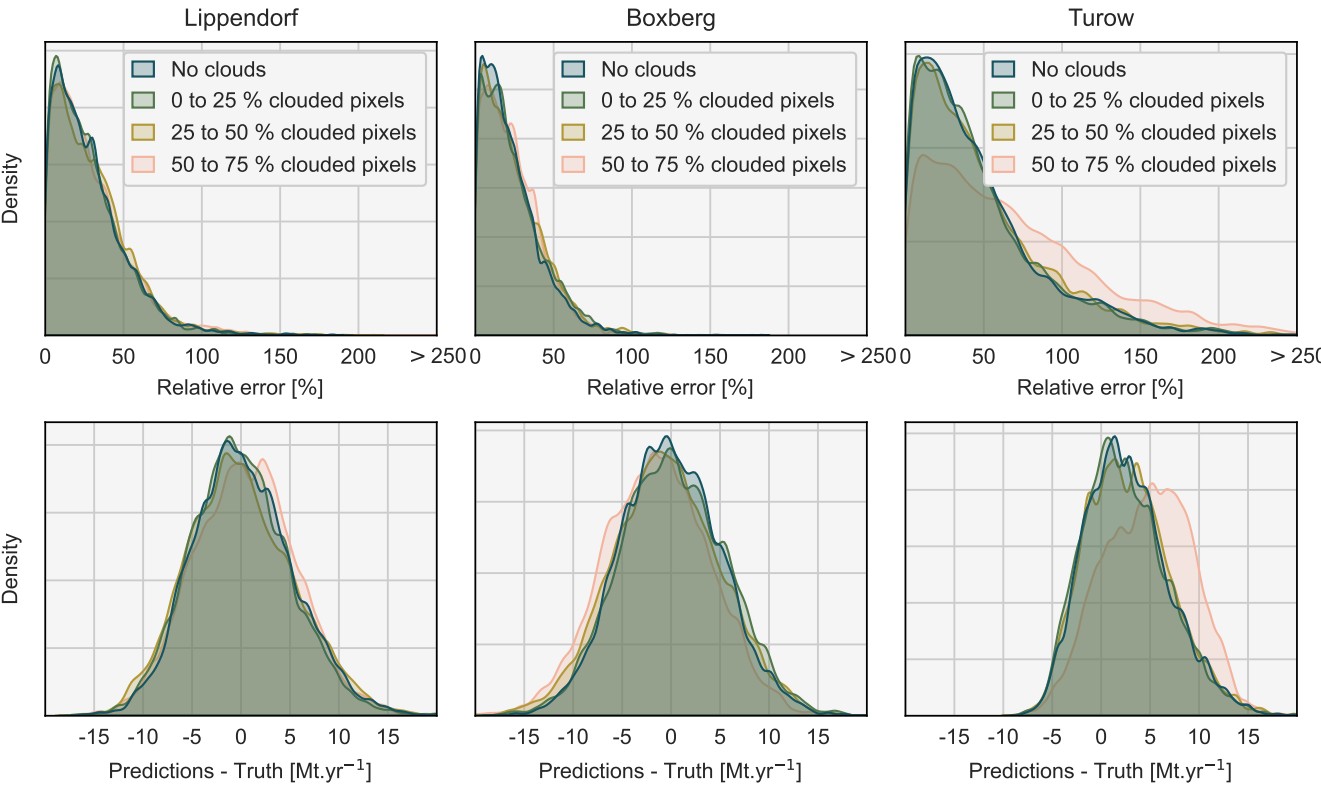

**Figure 6.** Density plots of the absolute relative error and of the algebraic difference between the CNN predictions and the reported emissions. The CNN models are trained and evaluated with $XCO_2$, wind, and time input data, affected by varying levels of cloud coverage. Predictions with absolute relative errors greater than $250\%$ or absolute errors greater than $15\,\mathrm{Mt.yr}^{-1}$ were set to 250 or 15 to increase visibility.

from $23.1\%$ to $18.5\%$. Finally, incorporating time or wind as a feature yields no significant benefit in the performance of the CNNs.

Concerning the influence of clouds, in the cases of Lippendorf and Boxberg, the accuracy of plume emission predictions is
not significantly compromised by their introduction, even with high cloud coverage exceeding 50%. This observation is valid whether or not $NO_2$ is factored into the analysis. However, for Turow, a power plant with lower emissions, the performance of CNN predictions degrades progressively with an increase in cloud coverage, notably when cloud coverage exceeds 50%. The specific decline in prediction accuracy for Turow can likely be traced back to the fact that the Turow's plume is mostly indistinguishable from the background. Consequently, the CNN's capacity to accurately estimate Turow's emissions is inherently
based on limited information, even in the absence of clouds. The introduction of cloud cover exacerbates this issue by further diminishing the available information.



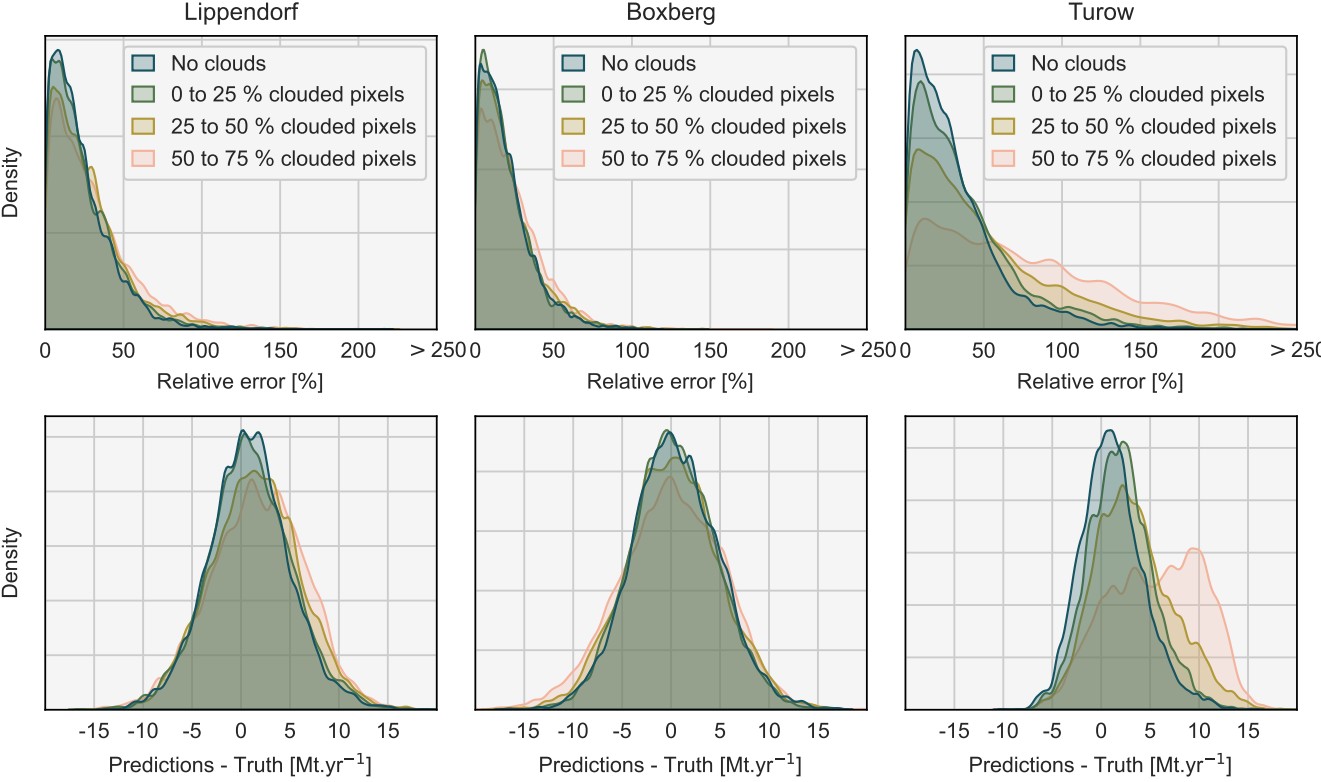

**Figure 7.** Density plots of the absolute relative error and of the algebraic difference between the CNN predictions and the reported emissions. The CNN models are trained and evaluated with $XCO_2$, wind, time, and $NO_2$ input data, affected by varying levels of cloud coverage. Predictions with absolute relative errors greater than 250% or absolute errors greater than $15\,\mathrm{Mt.yr}^{-1}$ were set to 250 or 15 to increase visibility.

## 4.2 Application to OCO-3 SAM observations

In this section, we assess the ability of CNNs trained on power plant plumes from the SMARTCARB synthetic dataset encompassing the power plants of Jänschwalde, Schwarze Pumpe, Boxberg, Lippendorf, Turow, Patnow, and Opole, to estimate emissions from real plumes observed at power plants by OCO-3 SAM, along with ERA5 wind fields and time information. A total of 39 observations of OCO-3 SAM data for 9 power plants are examined.

To obtain meaningful statistics from the small number of images, we use two different methods to increase the number of predictions for each image:

1. An ensemble of 100 images $x_i^1, ..., x_i^{100}$ for each normalised OCO-3 SAM observation $x_i$, where $x_i^j \sim \mathcal{N}(x_i, 0.3)$.

2. An ensemble of 16 neural networks, all trained with slightly different hyperparameters considering various levels of cloud coverage, and either uniform or beta distribution used for augmentation.





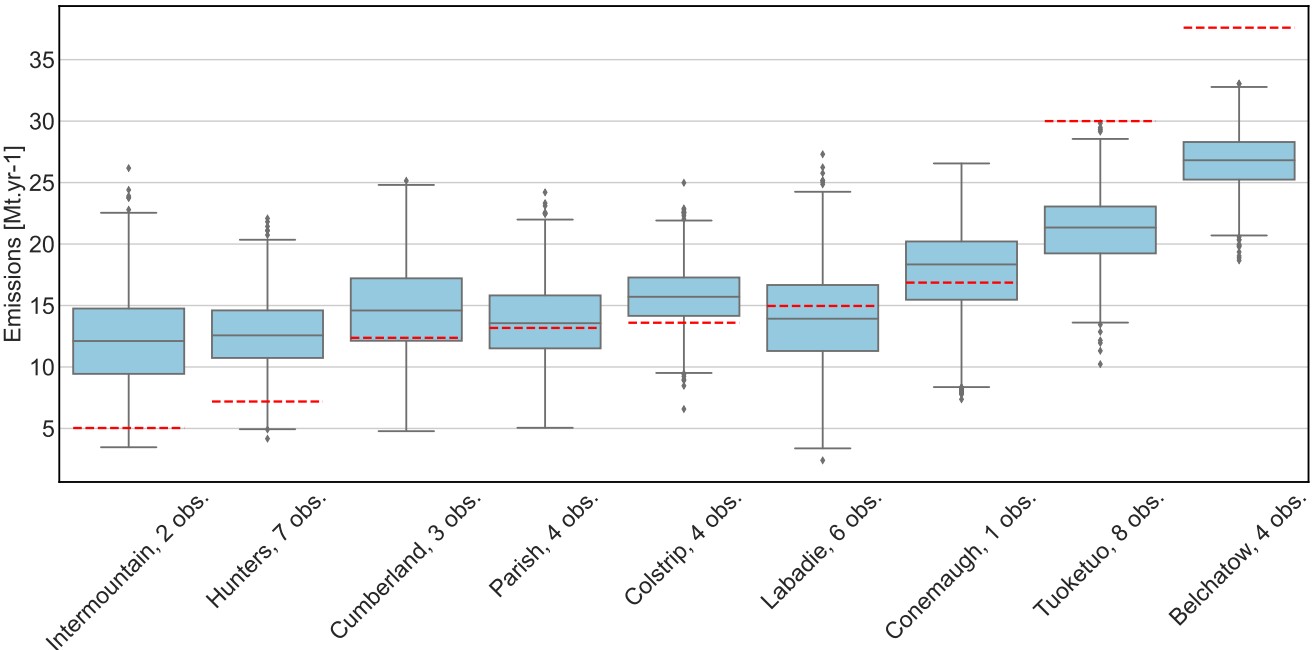

**Figure 8.** Boxplots of the ensembles of predictions based on the OCO-3 SAM observations for various power plants. Comparison with the reported annual emissions of the corresponding power plants (red dashed lines).

Each neural network generates 100 predictions from the 100 images. The ensemble mean should give a more accurate estimate than a single prediction for $x_i$ as seen with the synthetic data (see Section 4.1). Together with the 16 networks, we obtain 1600 predictions for each image, enhancing the robustness and reliability of our statistical analysis.

Figure 8 and Table 3 show the ensemble of predictions for each power plant compared to the annual reported emissions.

The median absolute and median absolute relative differences between the ensemble average predictions and the reported emissions are $7\,\mathrm{Mt.yr}^{-1}$ and 29%, respectively. This relative difference of 29% is only slightly higher than what was observed on synthetic satellite imagery. Specifically, CNNs exhibit a good match with reported emissions for power plants with emissions ranging between $10\,\mathrm{Mt.yr}^{-1}$ and $20\,\mathrm{Mt.yr}^{-1}$ (e.g., Colstrip, Cumberland, Labadie, Parish, Conemaugh). However, the

discrepancy with reported emissions largely increases for power plants at the extremes of low or high emissions (e.g., Belchatow, Tuoketuo, Hunters, Intermountain). Considering the 5% and 95% quantile predictions of Table 3, the emissions estimated by the CNNs range from $6.7\,\mathrm{Mt.yr}^{-1}$ to $30.4\,\mathrm{Mt.yr}^{-1}$, whereas Belchatow's reported emissions stand at $37.6\,\mathrm{Mt.yr}^{-1}$ and Intermountain's at $5\,\mathrm{Mt.yr}^{-1}$. This indicates that the variance in CNN predictions is significantly lower than that of the reported emissions. Given that the CNNs were trained on plumes with emission levels spanning from $0\,\mathrm{Mt.yr}^{-1}$ to $45\,\mathrm{Mt.yr}^{-1}$, it was

initially anticipated that they could accurately predict plumes akin to those from Belchatow or Intermountain. Furthermore, in Section 4.1, we show that the CNNs reliably recover the low emissions ($8.7\,\mathrm{Mt.yr}^{-1}$ in average) of Turow power plant.





| Power plant | Reported emissions | Predicted emissions | | | Location | | Number obs. |
| --- | --- | --- | --- | --- | --- | --- | --- |
| | | Mean | 5% | 95% | Latitude | Longitude | |
| Intermountain | 5.0 | 12.2 | 6.7 | 18.3 | 39.5 | -112.6 | 2 |
| Hunters | 7.2 | 12.7 | 8.2 | 17.5 | 39.2 | -111.0 | 7 |
| Cumberland | 12.4 | 14.7 | 8.6 | 20.7 | 36.4 | -87.7 | 3 |
| Parish | 13.2 | 13.6 | 8.7 | 18.6 | 29.5 | -95.6 | 4 |
| Colstrip | 13.6 | 15.7 | 12.0 | 19.6 | 45.9 | -106.6 | 4 |
| Labadie | 15.0 | 14.1 | 8.0 | 20.8 | 38.6 | -90.8 | 6 |
| Conemaugh | 16.9 | 17.6 | 9.3 | 22.2 | 40.4 | -79.1 | 1 |
| Tuoketuo | 29.5 | 21.2 | 16.4 | 25.9 | 40.2 | 111.4 | 8 |
| Belchatow | 37.6 | 26.7 | 22.9 | 30.4 | 51.3 | 19.3 | 4 |

**Table 3.** Annual reported emissions and statistics on the predicted emissions for the various power plants considered in the OCO-3 SAM constituted dataset. All emissions are in $\mathrm{Mt.yr}^{-1}$.

The subsequent analyses will explore the causes of the observed discrepancies between extreme reported emissions and CNN predictions.

Figure 9 shows the predictions of a randomly selected CNN model from the ensemble for 8 specific OCO-3 SAM images.
For each OCO-3 SAM image, we show a sensitivity map obtained by the integrated gradient method, which computes the gradient of the model's output (the emissions) relative to its input pixels, indicating how emissions are expected to increase or decrease with changes in pixel values. (see Dumont Le Brazidec et al. (2023) for details). Assuming that emissions estimates are directly correlated with the detection of plume pixels, the integrated gradient maps are anticipated to highlight a collection of positive pixels that effectively reconstruct the plume.

Images (a), (b), (c), and (d) are four instances of clear identification of the plume by the CNN. The integrated gradient method in each of these cases reveals collections of positive pixels forming a discernible plume shape. These positive pixels are encircled by negatives, suggesting that if the surrounding pixels intensified to match the plume's pixel values, the CNN would less likely recognise these as plume pixels, interpreting the aggregate as elevated background values instead. Predictions closely match reported emissions in each scenario, barring the anomaly of Intermountain. This discrepancy is logical, given
that the Intermountain plume is visually detectable in the image, whereas plumes corresponding to emissions of $5\mathrm{Mt.yr}^{-1}$ in the SMARTCARB dataset are typically obscured by the background.

Images (e) and (f) illustrate scenarios where clouds obscure the central portion of the image, thereby concealing a major part of the plume. These examples allow us to investigate how the CNN adapts to such conditions, making inferences based on the limited information available. In image (e), the CNN identifies a plume adjacent to the obscured area and bases its emissions
estimate on this collection of pixels. In image (f), the CNN interprets a significant cluster of high-value pixels as the tail end of the concealed plume and calculates emissions based on this inferred section of the plume.





**Figure 9.** Analysis of the predictions of a CNN (chosen randomly from the ensemble) on 8 specific OCO-3 SAM images. Each of the image is presented alongside the resulting map (on its right) from the application of the integrated gradient method and the reported and predicted emissions in $\mathrm{Mt.yr}^{-1}$. Power plant is indicated by a brown star.



**Figure 10.** Analysis of the predictions of a CNN (chosen randomly from the ensemble) for 3 high emission plumes. One plume from Belchatow and two from Tuoketuo (middle column) are compared against equivalent high emission plumes from the SMARTCARB dataset (left column). For each OCO-3 SAM based plume, the integrated gradient approach is applied and presented on the right column. To ensure a fair comparison between columns 1 and 2, identical colorbars have been used. Power plant is indicated by a brown star.

Images (g) and (h) shed light on a primary cause for the supposed overestimation of emissions from low-emitting power plants. These images feature barely discernible plumes alongside significant patterns (potentially systematic satellite errors) appearing on the left side of the images in both cases. The CNN mistakenly identifies these patterns as part of a plume in each case. Consequently, the model infers disproportionately high emissions based on this noise, leading to a substantial estimation of the emissions of these power plants.





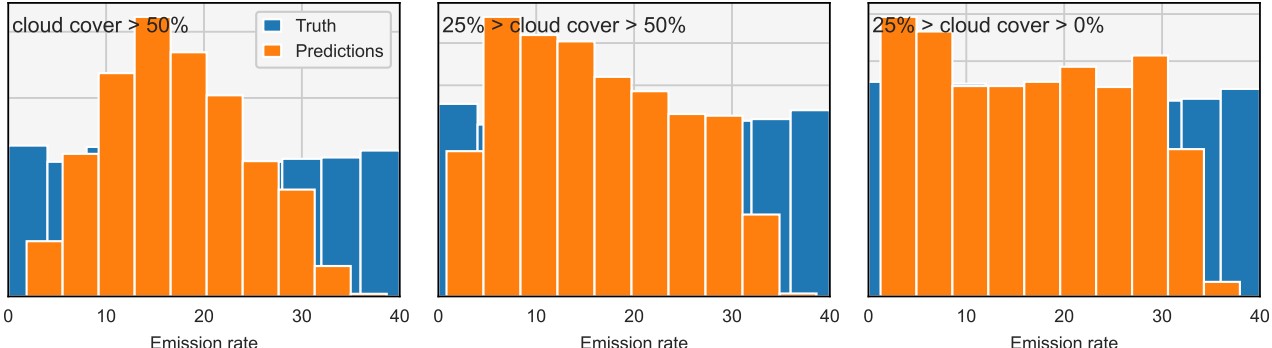

**Figure 11.** Distribution of the predictions of a CNN trained with uniformly distributed emissions, for synthetic images of the Dolna Odra power plant with uniformly distributed emissions, against the truth, at different levels of cloud coverage.

In Figure 10, we propose a first possible explanation for the underestimation of high emission plumes by the CNN. Three observed high emission plumes—one from Belchatow and two from Tuoketuo— are compared against SMARTCARB plumes at the Belchatow or Tuoketuo reported emission levels. Specifically, the SMARTCARB simulations are chosen to represent

emissions of 37 Mt.yr$^{-1}$ and 29.5 Mt.yr$^{-1}$, aligning with the reported emissions for Belchatow and Tuoketuo, respectively, and have comparable ERA5 wind speeds to those at Belchatow or Tuoketuo snapshot OCO3-SAM images. The simulated plumes appear more pronounced against the background than their real counterparts, suggesting a higher emission magnitude. This observation might account for the model's tendency to estimate lower emissions than Belchatow and Tuoketuo reported emissions. Further validation comes from the integrated gradient analysis, indicating accurate plume contour predictions by

the model and affirming that the relevant information was used for its estimations.

Another reason for the CNN underestimating high emission plumes could be regression towards the mean in scenarios with high cloud levels. To show this, we train a CNN with a dataset of power plants with uniformly distributed emissions between 0 and 40 Mt.yr$^{-1}$ and a low cloud coverage ($< 25\%$ of the image covered with clouds). In Fig. 10, we plot the distribution of the predictions of the CNN for synthetic images of the Dolna Odra power plant with uniformly distributed emissions, against

the truth, at different levels of cloud coverage. We observe a convergence to average values correlated with cloud coverage intensity. When the CNN lacks sufficient information in the image to infer emissions, it tends to average its predictions to minimise loss. A second observation is that even for low cloud coverage, the CNN struggles with emission levels higher than 33 Mt.yr$^{-1}$, while it is trained with emissions uniformly distributed between 0 and 40 Mt.yr$^{-1}$ (note that the CNNs trained in previous sections were trained for emission levels between 0 and 45 Mt.yr$^{-1}$). Increasing the number of high-emission plumes

in the training dataset would likely reduce the CNN's bias towards emissions near the upper limit defined in the training data.





## 5 Discussions and limitations

The ability of CNNs to estimate $CO_2$ emissions from power plants plumes was validated on synthetic satellite images. The presence of cloud coverage does not significantly affect the CNNs performance, except in instances of substantial cloud presence. CNNs demonstrate adaptability, leveraging residual information to accurately estimate emissions under heavily clouded

conditions. The inclusion of $NO_2$ data proves slightly beneficial, enhancing the CNN efficacy in all sky conditions.

The CNNs, once trained on simulated datasets of $XCO_2$ images, can be directly applied to real-world data with good accuracy. Nevertheless, it is observed that the spread of the CNN predictions is lower than the spread of the OCO3-SAM reported emissions. Predictions are significantly higher for low-emission power plants due to the presence of systematic errors in the image that are falsely identified as plumes and significantly lower for high-emission power plants. Furthermore, predictions

are lower than reported annual emissions for high-emission power plants. This is likely due to regression towards the mean in weakly informative images and/or discrepancies between the training and evaluation datasets. Finally, it is acknowledged that comparing instantaneous emissions measured during satellite overpasses with reports of annual average emissions from the EPA and E-PRTR inventories presents challenges, owing to the variability and intermittent nature of power production and $CO_2$ emissions.

The divergence between the distributions of real $XCO_2$ observations and those of the simulations observed in Section 4.2 necessitates CNN adaptation. To account for systematic satellite errors, a promising approach involves mingling real and simulated data during the training phase, such as overlaying a simulated plume of known emissions onto a real background. This method would introduce systematic errors typical of real satellite data while maintaining a controlled environment for supervised learning.

## 335 6 Conclusions and perspectives

In this paper, we improve the CNN model for the inversion of CO2 plumes from Dumont Le Brazidec et al. (2023) through the introduction of a novel data augmentation strategy and a dedicated approach to deal with clouds. This methodology was validated using the synthetic CO2M observations from the SMARTCARB dataset, demonstrating its efficacy in handling cloud-covered scenarios. Our findings indicate that, on average, clouds do not pose a significant challenge for CNNs, which maintain

high performance levels under both sparse and dense cloud conditions.An exception is observed in the case of the Turow power plant, where performance significantly drops. This decline is likely attributable to Turow's relatively low emission levels, which results in its plumes being inherently less distinguishable from the background.

Following its validation, the methodology is applied to OCO-3 SAM observations. In total 39 observations across 9 power plants, adjusted for resolution and shape to match CNN input requirements, are analysed. For each observation, an ensemble

of predictions is produced by CNNs trained on the SMARTCARB synthetic dataset. The results are promising, exhibiting a relative difference with the reported emissions only slightly superior to the relative error observed with the synthetic dataset. Specifically, predicted emissions for images from mid-level emission power plants, such as Colstrip and Parish, correspond very accurately to reported emissions. Moreover, through the application of integrated gradient techniques, it is demonstrated



that the CNNs effectively identify plumes in the OCO-3 SAM images and accurately estimate emissions from the plumes'
physical locations.

However, we observed that images capturing low and high emission power plants plumes are prone to overestimation and underestimation, respectively, in comparison to the reported emissions. Systematic satellite retrieval errors are identified as a frequent cause of overestimation in the low emission power plant images. These errors, often non-Gaussian and absent in the synthetic training dataset, lead to significant inaccuracies.

This study demonstrates the feasibility of applying neural networks to real satellite imagery of $XCO_2$ following training on simulated datasets. Although we advocate the integration of a hybrid training approach that incorporates both real and simulated images in order to improve the robustness and accuracy of the model, we provide a ready-to-use CNN $CO_2$ plume inversion tool based on satellite imagery.

*Data availability.* The datasets used in this paper are available on a compliant repository on https://doi.org/10.5281/zenodo.12788520 and
originate from https://zenodo.org/record/4048228. The weights of the CNNs are available on https://doi.org/10.5281/zenodo.12788520

*Author contributions.* JDLB contributed to conceptualisation, developed the methodology, implemented the software, conducted the investigation, performed formal analysis, created the visualisations, managed resources, and administered the project. PV contributed to the investigation and formal analysis. AF contributed to conceptualisation, methodology, and project administration. MB contributed to the conceptualisation, methodology, administered the project and secured funding. GB contributed to conceptualisation and methodology. GK
provided resources. JDLB wrote the original draft with GB, GK, AF, MB, and PV contributing by reviewing it.

*Competing interests.* The authors declare that they have no conflict of interest.

*Acknowledgements.* This project has been funded by the European Union's Horizon 2020 research and innovation programme under grant agreement N° 958927 (Prototype system for a Copernicus $CO_2$ service). CEREA is a member of Institut Pierre-Simon Laplace (IPSL).



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
