# Peer review of "Quantification of CO2 hotspot emissions from OCO-3 SAM CO2 satellite images using deep learning methods"

_Geoscientific Model Development, 2024_

## Author Response (AR1)

**Comments/questions of the reviewers are in blue.**

Answers of the authors are in black.

**First reviewer**

Joffrey et al. have developed a framework for applying CNN to estimate the power plant CO2 emissions from satellite imagery of the CO2 total column mixing ratios (XCO2). The model appears to be both resilient to common data challenges (e.g., cloud cover and limited satellite swath), and effective, with an initial validation on synthetic data from eight German power plants. Results from the application to real satellite data show a promising alignment with reported annual emissions, suggesting that the approach could be viable for broader CO2 monitoring efforts. The authors have done a commendable job of presenting the data. Overall, this study presents a compelling and innovative approach to monitoring CO2 emissions from power plants via satellite imagery, leveraging the power of deep learning. With some refinements in clarity and additional context around validation, this work has the potential to make a significant contribution to remote sensing and environmental monitoring fields.

Thank you very much for your comments and encouragement.

minor suggestion:

**Quantification of Model Performance**: Although the abstract mentions relative errors "close to 20%," it would be beneficial to briefly mention how these results compare to traditional or alternative inversion methods if applicable."""

We agree benchmarking is important. We have compared our method in depth with other methods in a previous paper on a simulated dataset (cite paper). We have added a more precise statement in the introduction of our manuscript about this work: "Our previous research (Dumont Le Brazidec et al., 2022) evaluated the models using only synthetic images, comparing them against light alternative methods where they demonstrated better performance with an absolute error about half of that of the cross-sectional flux method."

**Second reviewer**

Dumont Le Brazidec et al. introduced an innovative approach to quantifying CO2 emissions from atmospheric plumes observed in satellite imagery using Convolutional Neural Networks (CNNs). Drawing from a previous study conducted by the same team in 2023, the work represents a significant step in combining synthetic and real-world datasets to train models capable of reasonably estimating emissions, even under challenging conditions such as cloud cover or limited plume visibility. The study's topic is relevant and crucial given the challenges of extracting key information, such as local emission rates from remote satellite sensing. Integrating data augmentation, using ancillary data (e.g., NO2 and wind fields), and applying interpretability tools (e.g.,

integrated gradients) demonstrate methodological novelty and potential for real-world applications. In my opinion, while the study presents promising results and is well-aligned with the scope of **Geoscientific Model Development (GMD)**, certain aspects are yet to be improved for the manuscript to qualify for publication in the GMD. The revision suggestions, although multiple, qualify as "minor." Please refer to the comments below for more details.

We sincerely thank you for your thorough and constructive review of our manuscript.

**Technical Comments:**

I) On figures and their captions, in General: Most figures provide insufficient information on the units of the depicted physical property. The authors should clearly state the units, as they have done for Figures 1 and 3. Figures 9 and 10 are the most severe examples of this issue, where neither mixing ratios nor emission rates come with the proper units. Furthermore, multiple instances exist where the figure's details have not been appropriately explained in the caption. Note that since many readers will engage with the publication by checking out the figures, the captions need to be clear enough, even when clarifying explanations are mentioned in the text. For instance, in Figures 3, 9, and 10, the year to which the XCO2 map belongs should be noted. Also, in Figure 8, the outlier designation criterion should be clearly explained in the caption. In other words, the authors should be explicit about the percentiles to which the box plot whiskers correspond.

In the revised manuscript, we have made the following updates to the figures: Figure 3: Added the year 2015, corresponding to all SMARTCARB synthetic fields. Figure 4: Kept without physical units to avoid overloading the already dense visualization.

Figures 5-7: No changes needed as physical units were already present.

Figure 8: Explanations have been added in caption: "Box spans the quartiles (25th to 75th percentiles), whiskers extend to the last points within 1.5×IQR (interquartile range), and points beyond the whiskers represent outliers." Units were already provided. Figure 9: Clarification has been made about units:

Left plots: CO2 fields in ppmv

Right plots (integrated gradients): Specified these should be interpreted as dimensionless attribution scores rather than physical units Figure 10: ppmv units have been added.

**II) On subscript notation: The subscripts are managed inconsistently throughout the manuscript. For example, $XCO_2$ and $NO_2$ are also written as $XCO_2$ and $NO_2$ .**

Thank you for spotting this inconsistency. We have thoroughly checked and corrected them.

III) Citation system: The citation method needs proofreading. There are instances of

**putting the publication year within parentheses, while the parentheses have been omitted elsewhere.**

We are not sure to understand this statement. When citing other works, we use either \citep for parenthetical citations or \citet for in-text citations following the *natbib* rules, depending on the context, which are then processed by the copernicus.bst style.

IV) On the relation between Section 3.1.1 and Figure 4: This is another example where the reader should bounce back and forth between the text and the figure to understand the content properly. Although the layers in Figure 4 are for demonstration, it's still recommended to put color legends for the sake of completion. The noise layer lacks details, and the reader would not recognize it as a Gaussian noise with a given standard deviation unless reading the text. The text in Section 3.1.1 mentions augmenting the main data with either the beta or the uniform distribution; however, the scheme only shows the beta. The purple box in Figure 4 seems misleading. Rather than sequential execution, the relation between 4 and 5 is more characterized by the iterative nature of parameter adjustment during the optimization. A feedback arrow from 5 to 4 can better show the latter. This approach will reinforce the discussions in 3.1.5.

We agree that the figure's complexity could lead to confusion. To maintain readability while providing necessary context, we have modified the caption to clarify that synthetic satellite noise is modelled as gaussian noise with a standard deviation of 0.7 ppmv and zero mean. The caption now details our data processing steps: we scale major plumes using either beta or uniform distribution (with beta distribution shown here), modify the background with a uniform random field ranging from -3.5 to 3.5 ppmv, scale remaining plumes using a uniform distribution from 0.33 to 3, and mask pixels based on cloud coverage threshold. We have retained the color-coded boxes to distinguish between the model components (purple) and preprocessing layers (green).

V) Discussion around Figures 9 and 10, tortuous and difficult to follow: The panels of Figure 9 represent three conditions associated with panels (a)-(d), (e)-(f), and (g)-(h), discussed in the surrounding text. These panels are samples out of a bigger dataset. Why these eight and not the others? Have you inspected all the snapshots, discovered these three trends, and chosen to present these eight panels as "supporting examples"? If yes, please include some explanation in the text. Moreover, If you have examined other snapshots, aren't other significant plume trace-emission estimate trends to be presented? If you have only examined these eight snapshots, please advise why a broader inspection is not warranted if not required. I posted the comments above because I find it confusing to follow the overarching logical flow here, as I have difficulty understanding the natural order of evidence and conclusion. The discussion becomes even more dense and complex in lines 297 to 305. I see a complex issue being presented questionably. The logic is partially valid but could be problematic for the following reasons:

We have individually examined all 39 snapshots and identified the trends presented in the manuscript. We have added some explanation about this inspection process. "These images were chosen after a thorough inspection of all 39 snapshots in our dataset to illustrate the key patterns which we identified." (Section 4.2)

Beyond the patterns already discussed in the manuscript, no other significant trends were identified.

**1 Dependence on Background Contrast:**

• If the CNN interprets more pronounced plumes as higher emissions, it should perform well when the plume is distinct from the background. However:

• The authors argue that lower contrast in real plumes leads to underestimation.

• This implies that the CNN might not sufficiently generalize from simulated data to real-world conditions where plumes may naturally blend more with the background.

**2 Flaw in Justification:**

 The authors use the difference in plume-background contrast to explain underestimation. However, this reasoning partially undermines their earlier claims that CNN performs well on high-emission cases in synthetic data.

• The CNN was trained on synthetic images with high-emission plumes designed to be pronounced. If real-world high-emission plumes are less distinct, the model's training data might not adequately represent the variability in actual observations, leading to systematic bias.

We agree with your remark but note that this had been already acknowledged in the manuscript. In the discussion and limitations section, we speculate that systematic bias between OCO3-SAM observations and synthetic dataset is one potential explanation of the underestimation of pronounced plumes.

"The divergence between the distributions of real XCO\$\_2\$ observations and those of the simulations observed in Section 4 necessitates CNN adaptation."

We agree in-depth, out of the scope of this paper, analysis would be required.

**3 Mismatch Between Training and Real Data:**

• If the CNN was trained to interpret pronounced plumes as high emissions, it might struggle when the plume signal is weaker (as with actual data). This is consistent with their explanation but also highlights a limitation in their approach: a lack of robustness to discrepancies between simulated and real-world data.

To summarize this long comment, the mismatch between simulated and real-world data, especially in plume-background contrast, likely leads to systematic errors in CNN. While the authors' reasoning is plausible, it would benefit from a more substantial acknowledgment of these limitations and a discussion of how to address them in future work.

The dataset was balanced with weak plumes (and therefore weak signals) and pronounced plumes (high emissions/signals). We used either the uniform distribution or

the beta distribution (as described in Figure 5) to construct a balanced training dataset.

We agree that expanding the dataset is crucial, particularly through incorporating real-world observations. A combination of simulated data and actual measurements from OCO-3 SAM or the upcoming CO2M mission would be ideal for training. This approach is outlined in our suggestions for future work in that section.

"To account for systematic satellite errors, a promising approach involves mingling real and simulated data during the training phase, such as overlaying a simulated plume of known emissions onto a real background. This method would introduce systematic errors typical of real satellite data while maintaining a controlled environment for supervised learning."

Following your concerns, we have insisted in the revised manuscript on the divergence between training and test datasets in the discussions and limitations section "The divergence between the distributions of real XCO\$\_2\$ observations and those of the simulations observed in Section \ref{appli:OCO-3}, particularly in terms of systematic satellite errors, creates a domain shift between training and test conditions that likely leads to systematic errors in CNN predictions, necessitating CNN adaptation."

**Specific comments:**

Line 56: This sentence, "In previous studies, the models have only been tested with synthetic images without missing data," needs a citation.

We have now added a citation and expanded the sentence

"Our previous research (Dumont Le Brazidec et al., 2022) evaluated the models using only synthetic images without missing data, comparing them against light alternative methods where they demonstrated better performance

Line 82, item ii: This explanation is ambiguous and begs further clarification. • The phrase "satellite swath constraints" could refer to:

- The limited spatial extent of a single swath.
- Gaps in coverage between consecutive swaths.
- Temporal constraints where a given area is not revisited frequently.

•A reader unfamiliar with OCO-3 or the specific satellite mission might be left uncertain about what drives the need for a finer grid.

**We added the sentence**

"ii) satellite swath limitations - the limited spatial extent of a swath and temporal constraints between two swaths makes it unlikely that satellite imaging will consistently capture 128 km2 areas centred over emission sources"

Table 1: Please indicate the year(s) for which the data in the third column correspond. If multiple years are involved, include information on data variability.

We have added "The data spans from 2020 to 2023.".

**Line 116: Please provide exemplifying instances for "... this approach has limitations."**

We have added "and that the observation information might not be perfectly conserved".

The squares in Figure 3: Orange and red shades look too similar here. Why not show one square with dashed sides and the other with solid sides?

We increased the thickness of the sides to make it clearer.

Lines 158 to 160: The statement "either uniform or beta" is unclear. Please elaborate more. Is the choice between beta and uniform completely random? Figure 5 shows that the choice potentially leads to widely different coefficients, which is not a problem as you are trying to synthesize a vast artificial input database to be passed to the CNN but still begs questions about the details of the procedure.

We systematically tested both distribution options by training separate CNN models with either beta or uniform distributions. While we found that both approaches significantly improved performance compared to training without augmentation, there were no clear systematic differences between models trained with either distribution choice. We have clarified this in the text by adding "We train separate CNN models for each distribution choice."

**Line 177: "... ranging from 0.75 to 2..." Is the sampling from a uniform distribution?**

Yes exactly. This has now been mentioned in-text. Thanks for this valuable input.

Table 2: For clarity, mention that the table entries are percentiles.

We added "Entries in the table represent relative errors expressed as percentages of the true emissions."

Lines 237 and 238: I find this approach to presenting results awkward. Usually, when some factors do not turn out to be influencing, this condition is described in the text, and the graphs are plotted without including them in the procedure. However, one can still post the graphs, including the minor factors, in the paper's supplementary material. For your choice of manuscript configuration, I suggest showing the plots without the wind/date effect while mentioning this point in the text and the figure captions.

The inclusion of wind fields and time does not significantly improve performance but does not degrade it either. Since most experiments were conducted with these inputs, re-running them without would entail substantial computational costs without impacting the results.

Lines 239 and 240: It would illustrate your quantitative criterion for calling a difference significant. A difference of < 5% seems to have been deemed significant enough to be

discussed in lines 235 to 238. Figures 6 and 7 show values as low as the number below for Lippendorf and Boxberg across the mentioned scenarios of cloud coverage.

We are not sure to understand this comment: Figure 6 and 7 show that the distributions for Lippendorf and Boxberg and different scenarios of cloud coverage are more or less equal. The differences in relative error mean are inferior to 5%.

**Line 250: Still confused about whether wind fields and time information are required for the algorithm to function adequately.**

We did not evaluate whether wind and time information had an impact in the case of inverting OCO3-SAM images. From our analysis on the synthetic dataset, we can expect it does not have significant impact and that the algorithm should function without.

Table 3: I recommend removing this table as redundant because Table 1 and Figure 8 already address all the information it offers.

We agree with this comment. We therefore removed Table 3 and eliminated all references to it. Thank you.

Lines 280 to 289: This is improperly referring to a numbered item. Please consider using Figure 9(a), Figure 9(b), and so on instead of Images (a), (b), etc.

Indeed. We have updated the manuscript accordingly.

Section 6: The manuscript would benefit from more explicitly acknowledging its limitations and possible future directions. The authors could better emphasize how their approach improves upon traditional inversion methods beyond computational efficiency. In particular, while the manuscript includes references to related work, there is insufficient discussion on how this approach compares quantitatively with other lightweight or traditional inversion techniques regarding accuracy and computational efficiency. Moreover, the limited generalizability of the CNN to real-world data due to training biases is acknowledged but not addressed in-depth. Incorporating hybrid training datasets (real and synthetic) is suggested but not explored experimentally.

The key strength of our method lies in its enhanced capability to handle data with lower signal-to-noise ratios in CO2 plumes compared to traditional approaches, enabling us to analyse OCO3-SAM data which presents particular challenges for conventional methods.

**We have added the sentence**

"Once trained on simulated XCO2 images, the CNNs can be directly applied to real-world data with high accuracy, unlike traditional methods, which struggle to detect plumes and distinguish them from the background due to the low signal-to-noise ratio of CO2 plumes." The comparison between traditional methods and ML methods has been made on a synthetic dataset in a previous manuscript. We have mentioned this in the introduction as you suggested.

For OCO3-SAM data, direct comparison is difficult due to the low signal-to-noise ratio of the data and therefore the difficulty of applying traditional methods.

We agree that we do not incorporate hybrid training, although it appears to be the solution to solve the discrepancy between the training and the test dataset, but this is outside the scope of this manuscript.

Thanks again to the reviewer for her/his numerous valuable contributions and comments.

Citation: https://doi.org/10.519